# Brain-specific homeobox Bsx specifies identity of pineal gland between serially homologous photoreceptive organs in zebrafish

Hiroaki Mano [1], Yoichi Asaoka [1], Daisuke Kojima [1]* & Yoshitaka Fukada [1]*

The pineal gland functioning as a photoreceptive organ in non-mammalian species is a serial homolog of the retina. Here we found that Brain-specific homeobox (Bsx) is a key regulator conferring individuality on the pineal gland between the two serially homologous photo-receptive organs in zebrafish. Bsx knock-down impaired the pineal development with reduced expression of *exorh*, the pineal-specific gene responsible for the photoreception, whereas it induced ectopic expression of *rho*, a retina-specific gene, in the pineal gland. Bsx remarkably transactivated the *exorh* promoter in combination with Otx5, but not with Crx, through its binding to distinct subtypes of PIRE, a DNA *cis*-element driving Crx/Otx-dependent pineal-specific gene expression. These results demonstrate that the identity of pineal photoreceptive neurons is determined by the combinatorial code of Bsx and Otx5, the former confers the pineal specificity at the tissue level and the latter determines the photoreceptor specificity at the cellular level.

[1] Department of Biological Sciences, School of Science, The University of Tokyo, 7-3-1 Hongo, Bunkyo-ku, Tokyo 113-0033, Japan. *email: sdkojima@mail.ecc.u-tokyo.ac.jp; sfukada@mail.ecc.u-tokyo.ac.jp

Duplication of a biological structure is found widely in a body plan of various organisms. Such intraspecific similarity is found, for example, in forelimbs and hindlimbs of tetrapods[1], appendages of arthropods[2], teeth of mammals[3], and eyespots on butterfly wings[4], all referred to as serial homology[5–7]. Serially homologous structures share a number of morphological and molecular features to undertake similar functions, but, on the other hand, each of the structures shows a distinctive property that often relates to different adaptive roles. Elucidating the genetic basis for the similarity and the difference between the serial homologs thus provides a clue to understanding the mechanism of how evolutionary novelty is acquired.

The pineal gland in the vertebrate brain is an endocrine organ that synthesizes and secretes melatonin with circadian rhythmicity[8,9] and it functions as an extra-ocular photoreceptive organ in non-mammalian vertebrates[10]. Pineal photoreceptor cells are morphologically similar to retinal rod and cone photoreceptor cells in that the pineal and retinal photoreceptor cells possess a lamellar outer segment for efficient capture of photons[10]. A striking similarity between them is also found in molecular components of the phototransduction, such as opsin[11–13], G-protein transducin[14,15], cGMP-phosphodiesterase[16], cyclic nucleotide-gated cation channel[17], and interphotoreceptor retinoid-binding protein[15,18]. These morphological, physiological and molecular similarities indicate that the pineal and retinal photoreceptor cells are serial homologs, despite their discrete origins in neural development[19].

The gross similarity of these homologs appears to be regulated by Crx/Otx-family homeodomain transcription factors. Crx was first identified as a key regulator for terminal differentiation of the retinal photoreceptor cells in mice[20,21], and it directs gene expression in the pinealocytes as well[20,22]. Crx gene is directly transactivated by Otx2, another Crx/Otx-family member, in vitro[23]. Consistently, Otx2 serves as the pleiotropic role both in the pineal gland and the retina. Conditional knockout of mouse Otx2 in a photoreceptor cell lineage-specific manner induced cell-fate conversion of the photoreceptor cells into amacrine-like neurons in the retina and caused complete ablation of the pinealocytes in the pineal gland[23]. These results demonstrate that a common genetic network composed of Crx/Otx-family members plays a key role in both retinal and pineal photoreceptor cells. In parallel with these genetic studies, an in vitro analysis of cis-acting DNA elements suggested that pineal regulatory element (PIRE) with a consensus sequence of TAATC/T mediates the expression of several pineal genes such as arylalkylamine-N-acetyltransferase (AANAT), hydroxyindole-O-methyltransferase (HIOMT), and pineal night-specific ATPase (PINA)[24]. PIRE serves as a recognition sequence of Crx[24], which preferentially binds to a TAATCC motif and its derivatives[20,21], further supporting crucial roles of Crx/Otx-family proteins not only in the retina but also in the pineal gland.

In contrast to these advances in understanding the genetic basis for the similarity between pineal and retinal photoreceptor cells, little is known about mechanisms underlying their individuality. In particular, the mechanism specific to the pineal gland remains mostly elusive, although the presence of pineal photoreceptor-specific genes, such as chicken pinopsin[11,12] and zebrafish extra-ocular rhodopsin (exo-rhodopsin or exorh)[13], suggests a solid genetic basis for the pineal specificity. In the zebrafish, Floating head (Flh) and Otx5 contribute to developmental and homeostatic gene expression in the pineal gland[18,25]. Flh is a homeodomain transcription factor that plays a pivotal role in early organogenesis of the pineal gland[25]. However, flh is only transiently expressed during the pineal development[26] and therefore unlikely to participate in pineal-specific gene regulation at later stages. Zebrafish otx5 and crx are the two duplicated homologs of

mammalian Crx gene[27]. In contrast to the pleiotropic roles of Crx in mice, zebrafish Crx and Otx5 appear to function separately in the retina and the pineal gland, respectively[15,18]. However, this functional partitioning apparently contradicts the fact that crx and otx5 are expressed in both the pineal gland and the retina[15,18]. These observations, along with the restricted phylogenetic distribution of the duplicated crx/otx5 genes in the ray-finned fish lineage[27], suggest the presence of unidentified factor(s) responsible for pineal-specific development and gene expression.

We previously demonstrated that the zebrafish has two closely related homologs of rhodopsin gene, rhodopsin (rho) and exorh, which are expressed in retinal rod and pineal photoreceptor cells, respectively[13]. Using the promoter regions of these genes, we generated transgenic lines Tg(rho:egfp) and Tg(exorh:egfp), in which EGFP reporter is specifically expressed in retinal rod and pineal photoreceptor cells, respectively[28]. In the present study, we compared gene expression profiles between the pineal and retinal photoreceptor cells isolated by fluorescence-activated cell sorting and found selective expression of Brain-specific homeobox (bsx) in the pineal photoreceptor cells. Knockdown of Bsx led to down-regulation of pineal-specific exorh gene and also upregulation of retinal rho gene in the pineal gland. We found that Bsx directly binds to pineal-specific exorh promoter through PIRE motifs and transactivates it in combination with Otx5. These data demonstrate that Bsx is a key regulator conferring individuality on the pineal gland between the two serially homologous photoreceptive organs in vertebrates.

## Results

**Identification of Bsx as a pineal transcription factor**. We isolated EGFP-positive retinal rod and pineal photoreceptor cells by using fluorescence-activated cell sorting from the Tg(rho:egfp) and Tg(exorh:egfp) transgenic zebrafish lines, respectively (Supplementary Fig. 1a–i). Comparison of their gene expression profiles by ordered differential display method[29,30] revealed two pineal photoreceptor-specific genes encoding transcription factors, brain-specific homeobox (bsx) and insulinoma-associated 1a (insm1a) (Supplementary Fig. 1j, k).

Whole-mount in situ hybridization analysis of zebrafish larvae demonstrated that bsx is expressed in several brain regions including the pineal gland and the hypothalamus (Supplementary Fig. 2), an expression pattern similar to that of mouse Bsx, which is localized to the pineal gland, telencephalic septum, mammillary bodies and arcuate nucleus in the hypothalamus[31]. The zebrafish pineal gland is composed of two major types of neurons: photoreceptor cells and projection neurons, the latter innervating other areas of the brain[25]. At larval stages, exorh-positive photoreceptor cells occupy the dorsomedial portion of the pineal gland (Fig. 1a, f), while the projection neurons expressing pax6[25] locate laterally in the gland (Fig. 1b, g). bsx transcripts were detected in both medial and lateral regions of the pineal gland (Fig. 1c, h), indicating that bsx is expressed in both photoreceptor cells and projection neurons. Expression of bsx was also detected in the parapineal organ, which is located unilaterally to the left-side of the pineal gland (Fig. 1c, h, arrowhead). On the other hand, insm1a transcripts were detected widely in the developing central nervous system (Supplementary Fig. 2), and therefore we focused on bsx functions.

**Bsx depletion affects pineal development and differentiation.** The role of bsx in pineal development was assessed by knocking down bsx function with morpholino antisense oligo (MO) (Fig. 2). Injection of bsx MO induced no gross morphological abnormality in the body (Fig. 2c) but caused significant reduction in size of the larval pineal gland at 3.5 dpf (Fig. 2d, e). The size

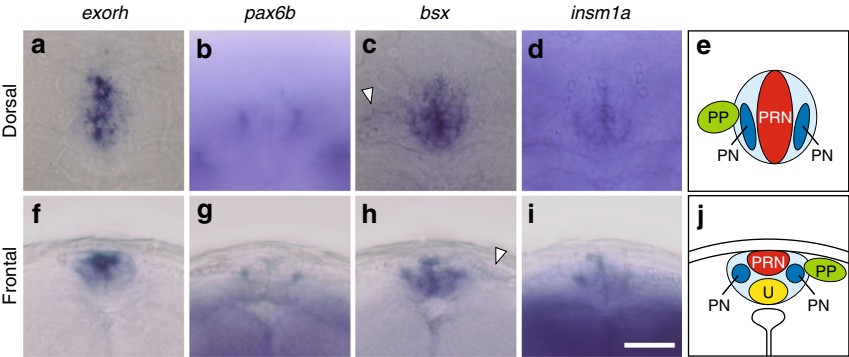

**Fig. 1** Expression patterns of *exorh, pax6b, bsx* and *insm1a* in the zebrafish pineal gland at 48 hpf. Schematic representations of the pineal complex are also shown in (**e**) and (**j**). Dorsal views (**a–e**) are oriented with anterior to the top, while frontal views (**f–j**) are with dorsal up. White arrowheads indicate *bsx* expression in the parapineal organ (**c**, **h**). Scale bar, 30 μm; PRN, photoreceptor cells; PN, projection neurons; PP, parapineal organ; U, undifferentiated/progenitor cells

reduction of the pineal body did not recover at later developmental stages (9.5 dpf; Fig. 2d, e), indicating that the *bsx* morphant exhibited a defect in pineal organogenesis rather than a transient delay of the pineal development. In addition, knockdown of *bsx* often led to emergence of pigmented cells within the pineal gland (Supplementary Fig. 3) and caused invasion of dermal melanophores onto its dorsal surface (Supplementary Fig. 3e, f), which normally forms a translucent pineal window[32,33]. These observations indicate a developmental role of *bsx* facilitating efficient light transmission to the pineal photoreceptor cells.

To investigate the role of Bsx in pineal-specific differentiation of photoreceptor cells, Bsx was depleted in *Tg(exorh:egfp)* or *Tg(rho:egfp)* transgenic line. Knockdown of Bsx reduced EGFP fluorescence signals in the pineal gland of *Tg(exorh:egfp)* larvae (Fig. 2f, g). The fluorescence intensity decreased to about 10% of the control at 3.5 dpf and slightly recovered at later stages (to ~25% and ~27% at 6.5 and 9.5 dpf, respectively) (Fig. 2g). Furthermore, Bsx depletion induced ectopic EGFP expression in the pineal gland of *Tg(rho:egfp)* larvae (Fig. 2h, i), in which EGFP fluorescence is normally detected only in retinal rod photoreceptor cells[28]. The ectopic EGFP expression was detected in all the larvae examined (*n* = 21) at 3.5 and 6.5 dpf but markedly decreased at 9.5 dpf (Fig. 2h, i). In *Tg(exorh:egfp);Tg(rho:ntr-mCherry)* double transgenic larvae (Fig. 2j), Bsx depletion similarly induced ectopic expression of the *rho:ntr-mCherry* transgene in a subset of pineal neurons, which were mutually exclusive from those retaining the *exorh:egfp* transgene expression (Fig. 2k). Consistently, whole-mount in situ hybridization of *bsx* morphants revealed severe reduction of endogenous *exorh* transcripts (Fig. 3a, f) and contrasting emergence of *rho* transcripts in the pineal gland (Fig. 3b, g). These effects of Bsx depletion were also confirmed in a *bsx* knockout mutant (*bsx*[m1376] line[34]; Supplementary Figs. 4 and 5) and by using another MO (*bsx* e1i1 MO, Supplementary Fig. 6). These results raise the possibility that Bsx plays a commanding role in determination and/or differentiation of the pineal photoreceptor neurons by specifying them differently from the retinal counterparts.

**Bsx functions upstream of Otx5 and downstream of Flh.** Transcriptional network for the pineal development was studied by examining genetic interactions between Bsx and other transcription factors. We found that pineal expression of *otx5* and *pax6b* (Fig. 3d, e) was downregulated by Bsx depletion (Fig. 3i, j), suggesting that Bsx functions genetically upstream of these factors. In contrast, Bsx knockdown had no marked effect on its own

transcript level in the pineal gland (Fig. 3c, h), while the *bsx*-positive area was reduced, consistently with the reduction of the pineal size (Fig. 2d, e). MO-based knockdown of Otx5 severely reduced *exorh* expression (Fig. 3k) with a marginal effect on *bsx* expression in the pineal gland (Fig. 3m), further supporting the idea that Bsx functions upstream of Otx5. Knockdown of Flh resulted in almost complete elimination of *exorh* and *bsx* signals at the presumed position of the pineal gland (Fig. 3n, p), in line with the pivotal role of Flh for pineal organogenesis[25], whereas Bsx depletion had no significant effect (by *bsx* knockout) or only a marginal effect (by MO-based knockdown) on *flh* expression at 48 hpf (Supplementary Fig. 5c, f). The results indicate that Bsx acts as a mediator downstream of Flh signaling in the developing pineal gland. On the other hand, substantial *bsx* expression remained in the parapineal organ of *flh* morphant (Fig. 3p, arrowhead). This observation is consistent with a previous report that the parapineal organ develops normally in *flh* mutant[35] and suggests that *bsx* expression in the parapineal organ is regulated in a manner independent of Flh activity. Unlike *bsx* morphant, neither *otx5* nor *flh* morphant induced *rho* expression in the pineal gland (Fig. 3l, o), emphasizing the unique role of Bsx in establishing the pineal specificity.

**Bsx transactivates *exorh* promoter in cooperation with Otx5.** To examine whether Bsx directly controls pineal-specific gene expression, we conducted a luciferase assay with a reporter construct harboring a 147-bp fragment of *exorh* promoter, which can direct pineal-specific gene expression in vivo[28]. Although Bsx alone had little effect on the promoter activity (Fig. 4a), co-expression of Bsx with Otx5, a known regulator for pineal gene expression[18], activated the transcription by 50-fold (Fig. 4a). In contrast, no synergistic action was observed when Otx5 was replaced by Crx (Fig. 4a), which mainly participates in regulation of retinal gene expression[15,18]. These observations suggest that the combinatorial action of Bsx and Otx5 serves as a definitive cue for the pineal photoreceptor-specific gene expression. This idea was further examined by ectopic expression of *bsx* in retinal photoreceptor cells, which intrinsically express *otx5*[18] but not *bsx* (Supplementary Fig. 2). Injection of *rho:bsx* construct into *Tg(exorh:egfp)* embryos induced EGFP expression in the retinal cells (Fig. 4b, c). These results demonstrate that Bsx transactivates pineal photoreceptor-specific genes in combination with Otx5 existing in both pineal and retinal photoreceptor cells.

**Bsx binds to *exorh* promoter through PIRE motifs.** To delineate the mechanism of Bsx action, we determined DNA-binding specificity of Bsx by a selection and amplification binding (SAAB)

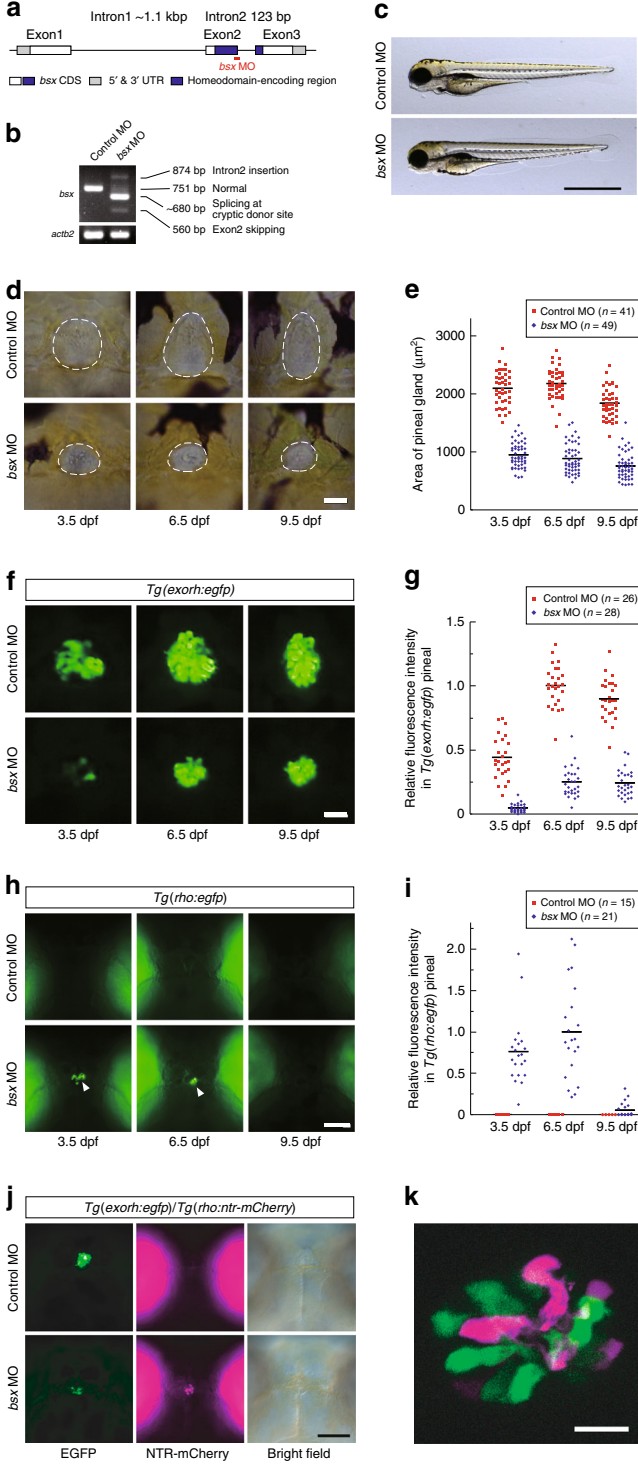

**Fig. 2** MO-mediated knockdown of *bsx*. **a** Schematic illustration of the target site of *bsx* MO. **b** RT-PCR of *bsx* mRNA in 48-hpf embryos injected with control or *bsx* MO. Full images of the electrophoreses are shown in Supplementary Fig. 8. **c** Apparent views of 3.5 dpf larvae injected with control or *bsx* MO. No gross morphological difference was detected. **d–k** Effects of *bsx* MO on the pineal gland. All the larvae are viewed dorsally with anterior to the top. Three images in each row (**d**, **f**, **h**) were taken from the same larva at different stages. **d** Size reduction of the pineal gland in *bsx* morphants. White dashed lines indicate the position of the pineal gland. **f** Reduction of pineal EGFP fluorescence in *Tg(exorh:egfp)* by Bsx depletion. **h** Induction of EGFP expression in *Tg(rho:egfp)* by Bsx depletion (indicated by white arrowheads). **e**, **g**, **i** Quantification of pineal size and fluorescence signals in *bsx* MO- or control MO-injected larvae. Horizontal bars indicate mean values. The difference between *bsx* MO-injected and control groups was statistically significant for each case [$P < 2.2 \times 10^{-16}$ (3.5, 6.5, and 9.5 dpf, **e**), $P = 2.8 \times 10^{-12}$ (3.5 dpf, **g**), $P < 2.2 \times 10^{-16}$ (6.5 and 9.5 dpf, **g**), $P = 2.0 \times 10^{-8}$ (3.5 dpf, **i**), $P = 1.3 \times 10^{-7}$ (6.5 dpf, **i**), $P = 0.011$ (9.5 dpf, **i**) by Welch's two-sided *t*-test]. **j** Effects of *bsx* MO on *Tg(exorh:egfp);Tg(rho:ntr-mCherry)* double reporter line at 5.5 dpf. In control larva, EGFP (green) and NTR-mCherry (magenta) were localized in the pineal gland and the retina, respectively. In *bsx* morphant, both EGFP (green) and NTR-mCherry (magenta) signals were detected in the pineal gland. **k** A confocal image of pineal cells in Bsx-depleted *Tg(exorh:egfp);Tg(rho:ntr-mCherry)*, showing mutually exclusive expression of EGFP (green) and NTR-mCherry (magenta). Scale bars, 1 mm (**c**), 30 μm (**d**, **f**), 100 μm (**h, j**), 10 μm (**k**)

GST-Bsx protein showed a binding activity to BSXRE probe (Fig. 5b), whereas no protein-DNA complex was detected with a mutated probe (Fig. 5b, BSXRE-mut probe). These results indicate that Bsx binds to the TAATCGGT motif.

Based on the matching scores with the SAAB matrix (Fig. 5a), we found six potential sites for Bsx binding (P1–P6, matching scores 5.43–6.35) in the 147-bp *exorh* promoter region (Fig. 5c, d). By performing a competitive EMSA using BSXRE probe, we found that P3 sequence remarkably competed with BSXRE for Bsx binding, while P1 and P4 had weaker but evident competitive activities (Fig. 5d). It should be noted that these three sites harbor TAAT core motif and the others do not (Fig. 5d), indicating that the TAAT core is essential for Bsx binding. The very low, if any, competitive activity of P2 (Fig. 5d) implies that PIPE (TGACCCCAATCT), a *cis*-acting element participating in pineal-specific gene expression[28], is less likely to serve as a Bsx-binding site. We further examined four TAAT-containing sequences that locate more distally in *exorh* upstream region (D1–D4 in Fig. 5c). D2, D3, and D4 had substantial Bsx-binding activities, whereas D1 showed a much lower activity despite harboring the TAAT core sequence (Fig. 5d).

EMSA using a probe of the 147-bp *exorh* promoter region (Fig. 5e) showed that Bsx directly interacts with the pineal-specific promoter (lanes 1 and 2). Bsx binding was remarkably reduced when P3 site was mutated, whereas mutations in P1 and P4 had only marginal effects on the formation of Bsx-bound complexes (Fig. 5e, lanes 3–5). Introduction of double mutations into P3 and P4 sites almost completely abrogated the Bsx-bound complex formation (Fig. 5e, lane 8). These results demonstrated that Bsx binds to the *exorh* promoter primarily through P3 and P4, both of which harbor the known *cis*-element PIRE. PIRE has long been considered as a binding site for Crx/Otx-family proteins, but the present data indicate that certain types of these pineal-specific *cis*-elements serve as binding sites for Bsx.

## Discussion

In the present study we identified Bsx as a regulator for pineal-specific cell differentiation and gene expression in the zebrafish.

assay using a glutathione *S*-transferase (GST)-Bsx fusion protein. Repetitive rounds of selection by GST pull down and PCR amplification revealed an octanucleotide motif, TAATCGGT, as a consensus sequence for Bsx binding (Fig. 5a and Supplementary Fig. 7). This sequence harbors a pineal regulatory element PIRE (TAATC/T), which has been characterized as a putative binding site for Crx/Otx[24]. To confirm the binding of Bsx to the sequence determined by SAAB, we performed an electrophoretic mobility shift assay (EMSA) with BSXRE probe, a synthetic oligonucleotide probe (36-mer) containing the highest matching sequence (Fig. 5b; matching score 9.54, see the legend for its definition).

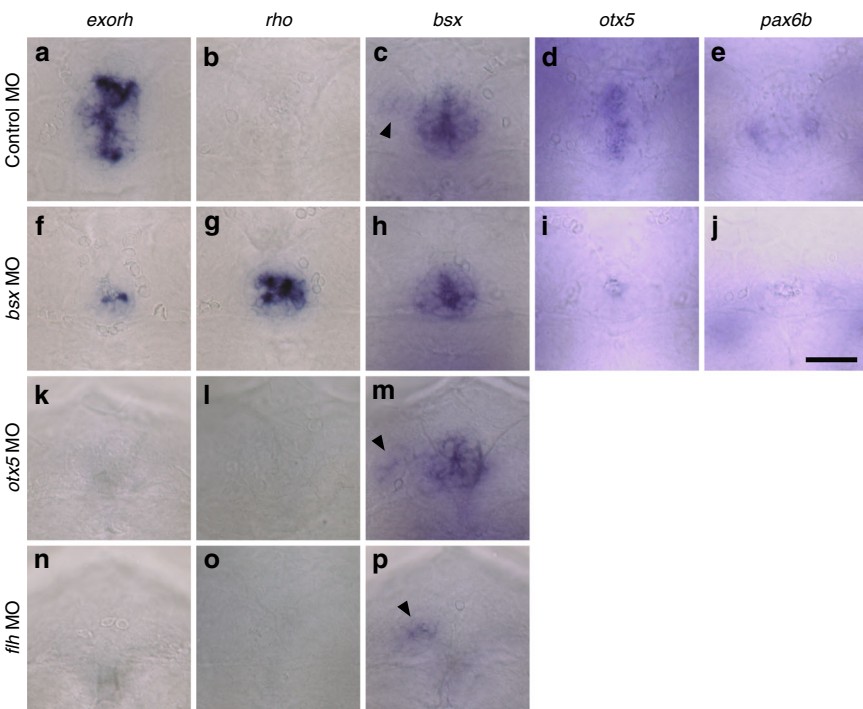

**Fig. 3 Effects of depletion of Bsx, Otx5 or Flh on the pineal gene expression at 48 hpf. a–p** Each panel represents head images of embryos ($n = 12$–32), all showing similar results. Embryos are viewed dorsally with anterior to the top. Arrowheads in (**c**), (**m**), and (**p**) indicate *bsx* expression in the parapineal organ. Scale bar, 30 µm

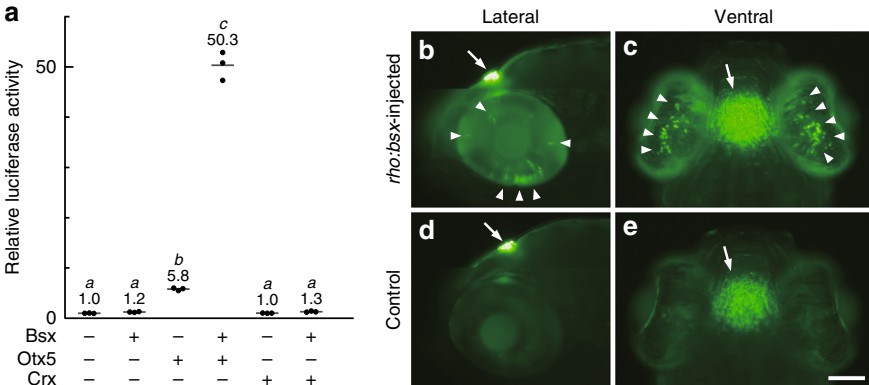

**Fig. 4 Bsx transactivates *exorh* promoter in vitro and in vivo. a** Cooperative transactivation of the 147-bp *exorh* promoter by Bsx and Otx5. Horizontal bars indicate mean values of relative luciferase activities ($n = 3$). Significant differences were observed between two groups that do not share the same lowercase letter ($P < 0.01$ by two-sided, Games-Howell test). **b–e** EGFP induction in the retina of *Tg(exorh:egfp)* fish by ectopic *bsx* expression under the control of *rho* promoter. Lateral (**b**, **d**) and ventral (**c**, **e**) views of *rho:bsx*-injected larva (**b**, **c**) or a control larva without injection (**d**, **e**) at 6.5 dpf. Arrowheads indicate GFP-positive retinal cells induced by *rho:bsx* injection, and arrows indicate pineal GFP fluorescence signals originally found in *Tg(exorh: egfp)* fish. Each of panels (**b**) and (**d**) is the montage of two photographs taken at different focal planes. Scale bar, 100 µm

Previous studies in mice[36], *Xenopus*[37] and, very recently, zebrafish[34] reported that Bsx regulates pineal development, but it remained unclear whether Bsx participates in establishing pineal-specific traits differently from retinal ones. We showed that Bsx depletion led to the ectopic differentiation of retinal-type photoreceptor cells in the pineal gland (Figs. 2h, k and 3g). In contrast, ectopic expression of Bsx in retinal photoreceptor cells upregulated pineal photoreceptor-specific *exorh* gene therein (Fig. 4). In both cases, pineal-retinal specificities of the cells were disturbed, whereas their identities as photoreceptor cells were maintained. Together with the finding that *bsx* is expressed in both pineal photoreceptor cells and projection neurons, the present results suggest that Bsx functions primarily to specify the pineal identity (Fig. 6a).

We demonstrated that Bsx remarkably activates the pineal-specific *exorh* promoter only when co-expressed with Otx5 (Fig. 4a). This synergistic effect provides an answer to a long-standing question about pineal-specific gene expression, which was not fully explained by Otx5 function alone. We propose a model in which pineal photoreceptor-specific gene expression is accomplished by the combinatorial action of two synergistic signals: Otx5 for a common framework of photoreceptor cells and Bsx for the individuality of the pineal gland (Fig. 6b).

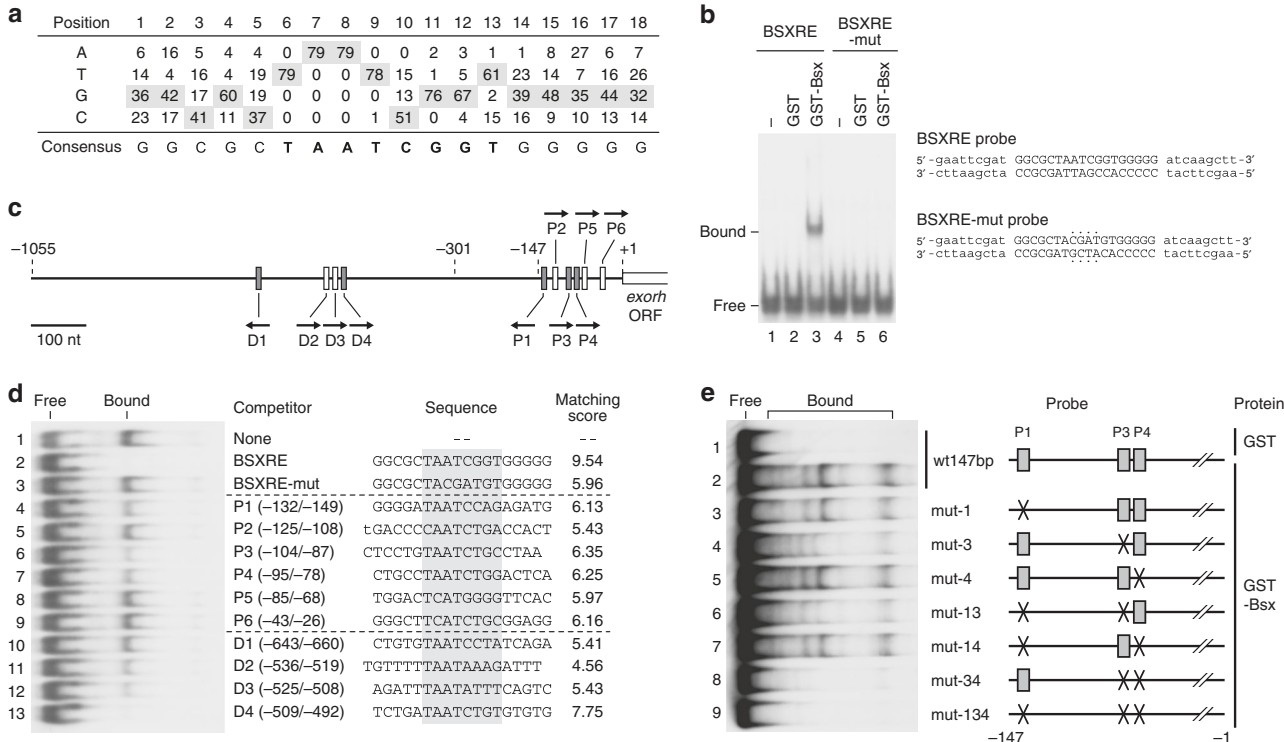

**Fig. 5** Bsx directly binds to *exorh* promoter via PIRE sites. **a** A position-specific score matrix of Bsx recognition sequences obtained by SAAB. See Supplementary Fig. 7 for raw sequence data. **b** EMSA with BSXRE probe in the presence of GST-Bsx protein. BSXRE probe harbors the 18-bp highest matching sequence determined by SAAB (BSXRE; indicated by capital letters), which is flanked by a pair of short arm sequences (indicated by lowercase letters). Mutated sequences in BSXRE-mut probe were highlighted with dots. **c** A schematic structure of zebrafish *exorh* promoter. Gray rectangles represent DNA elements matching the consensus sequence of PIRE (TAATC/T), whereas white ones represent other potential sites for Bsx binding. See (**d**) for the nucleotide sequences. Arrows indicate the direction of each element. Nucleotide positions are given relative to the translation initiation site. **d** Competitive EMSA against BSXRE probe. Each competitor oligonucleotide has an 18-bp sequence of interest, which is flanked with the arm sequences shown in (**b**). Matching scores are calculated as the sum of relative appearance frequencies of nucleotides using the score matrix in (**a**) at the positions 4 through 15. P2 oligonucleotide contains the 12-bp sequence of PIPE, in which the nucleotide at the 5′-most position overlaps the left arm sequence (indicated by a lowercase letter). **e** EMSA with *exorh* promoter sequence (wt147bp) or its mutated ones. Multiple shifted bands possibly represent the variety of higher order structures of protein-probe complex. Full images of the electrophoreses (**b**, **d**, **e**) are shown in Supplementary Fig. 8

Previous studies on homeodomains demonstrated that the amino acid at position 50 in the homeodomain participates in recognizing two nucleotides adjacent to the 3′ end of TAAT core DNA motif[38–40]. The homeodomains having glutamine at this position (Q50-type) generally recognize TAATTG, TAATGG, or TAATTA sequences, whereas those having lysine (K50-type) prefer TAATCC or its derived sequences[38–40]. Although zebrafish Bsx has glutamine at the position 50 (Q50-type), it preferentially binds to TAATCGGT (Fig. 5a), which is partially similar to the K50-type recognition sequence. Such an atypical property of Bsx protein allows it to bind to DNA *cis*-element PIRE (TAATC/T), which was characterized as a recognition sequence of Crx/Otx, K50-type homeodomain proteins[24]. The fact that PIRE is the common binding motif of both Bsx and Crx/Otx reasonably explains the reason why pineal-specific promoters/enhancers require multiple PIRE sequences for their activities[28,41]. As demonstrated in the competitive EMSA (Fig. 5d), Bsx had higher affinities to P3, P4, and D4 (TAATCTG motif) than to P1 and D1 (TAATCCT/A) among the five PIRE-containing sequences in the *exorh* promoter/enhancer (Fig. 5c, gray rectangles). These observations support a model in which the PIRE-containing sequences can be classified into two categories, Bsx- and Crx/Otx-binding sites, which together define pineal photoreceptor-specific gene expression (Fig. 6b). In fact, promoter/enhancer regions of several other pineal-specific genes possess potential Bsx

recognition sequences with high matching scores (Supplementary Table 1).

In the present study, Bsx knockdown by MO induced a number of pineal-specific phenotypes in larval zebrafish, such as the regressed pineal size (Fig. 2d), the reduced *otx5* expression (Fig. 3i) and the elevated *rho* expression (Figs. 2h, i and 3g). Consistently, most of these phenotypes have been reported in the *bsx* knockout mutant of zebrafish[34], indicating successful loss-of-function of Bsx by MO-based knockdown in our experiments. Bsx knockdown also resulted in the significant reduction of *exorh: egfp* transgene expression (Fig. 2f, g) as well as the marked decrease in the number of *exorh*-expressing photoreceptor cells in the larval pineal gland (Fig. 3). These *exorh* phenotypes of *bsx* morphants contrast with the previous study[34], which reported that *exorh* mRNA signals are upregulated in the pineal gland of *bsx* knockout mutant in an in situ hybridization experiment. Such an apparent difference in *exorh* expression level might be derived from different designs in the *exorh* cRNA probes of the in situ hybridization experiments. The previous study[34] employed an *exorh* probe for its protein-coding sequence, whereas we designed it for 3′ non-coding sequence unique to *exorh* mRNA (see the Methods section for details) to avoid possible cross-reaction to its closest homolog, *rho*. In fact, the *bsx* knockout mutant showed a reduced level of *exorh* expression not only in our in situ hybridization experiment (Supplementary Fig. 4) but also in the

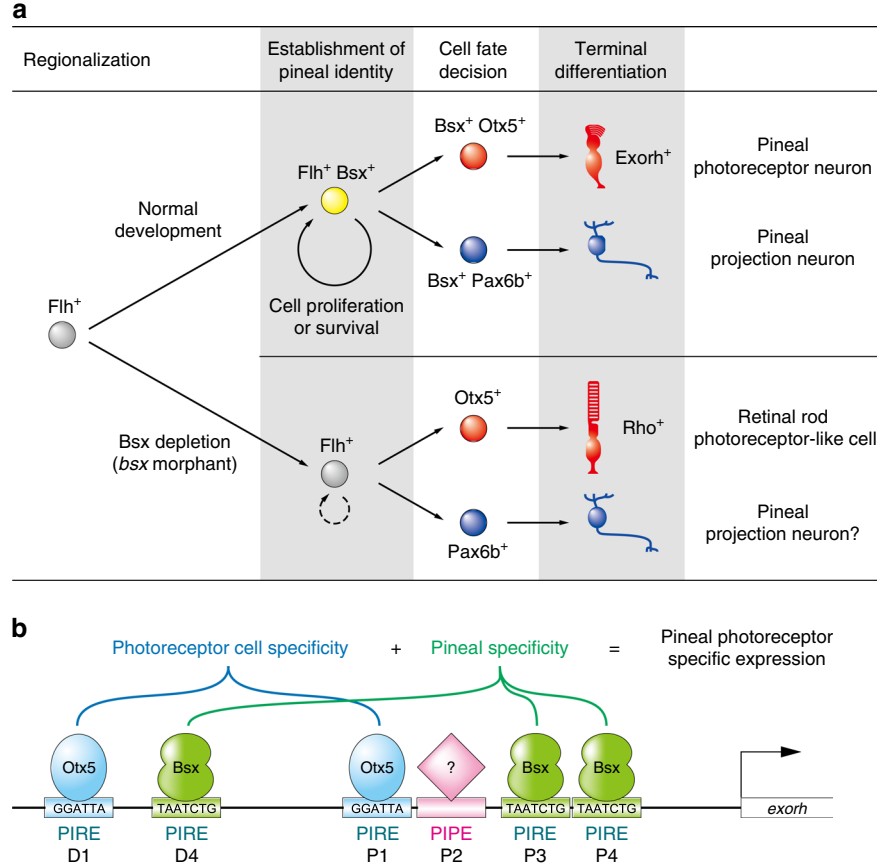

**Fig. 6** Models for specification mechanisms of the zebrafish pineal gland. **a** A model for Bsx function in the pineal development. **b** A model for transcriptional network regulating pineal photoreceptor-specific gene expression. The specificity is defined by the combinatorial action of Bsx (pineal specificity) and Otx5 (photoreceptor cell specificity) through multiple PIRE sequences, possibly in cooperation with another uncharacterized factor that binds to PIPE[28]

quantitative PCR experiment (Supplementary Fig. 5). These results are consistent with our cell-based promoter assay demonstrating as much as 50-fold transactivation of *exorh* promoter by Bsx in the presence of Otx5 (Fig. 4a).

Replication of a biological structure and the subsequent divergence in its evolutionary descendants generates serial homologs in individual organisms[5–7]. while the genetic mechanism directing serial homologs toward forming distinct structures has remained largely unknown. In this context, the pineal gland and the retina provide an excellent model, as they share a number of structural, functional and molecular similarities and also show remarkable differences in physiological function, cellular organization and gene expression. The present study demonstrates that zebrafish Bsx plays a pivotal role in pineal-specific cell differentiation between the two serially homologous tissues. Bsx regulates pineal-specific gene expression through a direct cooperation with Otx5, which contributes to similarity in gene expression between the two serially homologous photoreceptive tissues. The identification of Bsx should be an important milestone in the evolutionary developmental studies on the mechanisms of how the different neuronal tissues and cells have diverged functionally.

## Methods

**Fish strains**. RIKEN WT, EkkWill, and ABTL were used as wild-type strains. In the present study, Ex(-1055) transgenic line (*ja1Tg* strain), in which EGFP was driven by a 1055-bp fragment of *exorh* promoter[28], was renamed *Tg(exorh:egfp)* according to the zebrafish nomenclature guideline. Rh(-1084) transgenic line (*ja2Tg* strain), in which EGFP was driven by a 1084-bp fragment of *rho*

promoter[28], was also renamed *Tg(rho:egfp)*. The *Tg(rho:NTR-mCherry)* line (*ja83Tg* strain) was generated by introducing a transgene having the 1084-bp *rho* promoter[28] and the nitroreductase (NTR)-mCherry fusion gene[42]. The *bsx^m1376* mutant line[34] was a kind gift from Dr. Wolfgang Driever. All research described here adhered to local guidelines of the University of Tokyo, and all appropriate ethical approval and licenses were obtained from Institutional Animal Care and Use Committees of The University of Tokyo.

**Whole-mount in situ hybridization**. To minimize potential cross reactivities to homologous genes, digoxigenin (DIG)-labeled RNA probes (216–306 bases in length) were designed for 3′ UTRs of the zebrafish mRNAs for *exorh* (NM_131212.2, nucleotide 1112–1371), *rho* (AF109368, 1207–1422), *insm1a* (NM_205644.1, 1293–1588), *otx5* (NM_181331, 1300–1601) and *bsx* (GRCz11, chromosome 10, nucleotide 29,890,307–29,890,615). Because zebrafish *pax6b* mRNA has a short 3′ UTR, the *pax6b* probe was designed for the region covering 168 bases of the ORF and the following 134 bases of the 3′ UTR (NM_131641, 1407–1708). Each DNA fragment was cloned into the EcoRV site of a modified plasmid vector, termed pGEM15H, in which the multiple cloning sites of pGEM 3Zf(+) was substituted by the EcoRI/HindIII fragment excised from pBluescript II SK(+). The RNA probes were synthesized by using DIG RNA Labeling Kit (Roche Diagnostics).

Whole-mount in situ hybridization was performed as described previously[43]. Briefly, embryos at 48 or 78 hpf were fixed in 4% PFA in phosphate-buffered saline overnight at 4 °C and dehydrated in methanol. The samples were rehydrated and pre-treated with proteinase K and fixed again in 4% PFA in phosphate-buffered saline for 20 min at room temperature. The pre-treated larvae were hybridized with the DIG-labeled RNA probes, and the hybridization signals were visualized by staining with BM purple (Roche) or NBT/BCIP mixture. After staining, the specimens were immersed in 80% glycerol and examined with an upright microscope (Axioplan2, Zeiss).

**Microinjection of morpholino antisense oligos**. Sequences of morpholinos (MO) are as follows: *bsx* MO, 5′-ATTTA ACGCA ATTAC CTGTG TTTCC-3′; *bsx* e1i1 MO, 5′-TATAG GCTGC ACTTA CCAGA GGTGA-3′; *otx5* MO, 5′-CATGA

CTAAA CTCTC TCTCT CTCTC-3′; *flh* MO, 5′-AATCT GCATG GCGTC TGTTT AGTCC-3′. The *bsx* MO was synthesized by GeneTools (Philomath, USA). The *otx5* and *flh* MOs were obtained from Open Biosystems (Huntsville, USA). These MOs synthesized were dissolved in distilled water containing 0.05% phenol red at a concentration of 8 mg/ml (*bsx* MO), 4 mg/ml (*bsx* e1i1 MO) or 0.5 mg/ml (*otx5* and *flh* MOs). One-cell-stage embryos of wild-type or transgenic fish were injected with 2 nl of the MO solutions into the yolk immediately beneath the cell body. For control experiments, we used the standard control morpholino (Gene-Tools) in the same condition. The effects of the MO injection on *bsx* transcripts were evaluated at 48 hpf by RT-PCR with a pair of BSX3F (5′-GACGC ACGGA TTGTT CGC-3′) and BSX3R (5′-GTTGA TTTAG TGTAA TAATA GC-3′) primers. ZBAF1 and ZBAR2 primers[44] were used for RT-PCR of β-actin 2 (*actb2*).

**RNA extraction and RT-qPCR analysis**. For experiments using *bsx*^*m1376*^ mutants, embryos at 48 or 78 hpf were collected from crossing pairs of the heterozygotes, and dissected into anterior and posterior segments on ice; the posterior segments were used for genotyping of *bsx*^*m1376*^ allele[34], while the anterior segments were soaked in RNA*later* and stored at 4 °C. After genotyping, four or six of the anterior segments for each genotype were pooled as a biological replicate. For MO-injected embryos, four or six of the anterior segments were pooled as a biological replicate as well. Total RNA was extracted and purified with RNeasy Mini Kit (Qiagen). An equal amount of RNA from each sample was reverse-transcribed into cDNA with a mixture of the oligo (dT)$_{15}$ primer and the random primers with GoScript™ Reverse Transcriptase (Promega). The reverse-transcribed cDNA was then subjected to quantitative PCR using GoTaq qPCR Master Mix (Promega) and the StepOnePlus™ Real-time PCR system (Applied Biosystems) according to the manufacturers' protocols. Relative expression levels were calculated with a relative standard curve created with serially diluted cDNA samples of the 48-hpf anterior segments. The expression levels were normalized to β-actin 2 (*actb2*) expression levels in all the figures. The primers used for quantitative PCR are 5′-TGTG TTCAT GACG GTTCC TG-3′ and 5′-ACACC GTTGA CAACA TGCAG-3′ for *exorh*, 5′-ACAGG CTATC CCGTG TTCTG-3′ and 5′-GCCAG TACTG CATCT TGTGC-3′ for *flh*, 5′-TCCGA GACCA CACAG CG-3′ and 5′-CTGCT TGTTC ATGCA GATG-3′ for *rho*, and 5′-GGCAA TGAGA GGTTC AGGTG-3′ and 5′-GTGGT ACCAC CAGAC AATAC-3′ for *actb2*.

**Preparation and injection of *rho:bsx***. A full-length coding sequence of *bsx* was obtained by RT-PCR from adult zebrafish pineal gland cDNA with a pair of primers, BSX3F and BSX3R, and cloned into pCR2.1-TOPO vector (Invitrogen). To generate an expression construct *rho:bsx*, the coding sequence of *bsx* was PCR-amplified from the cDNA plasmid with primers, 5′-CAAGC TTCAT GAATC TGAAC TACAC G-3′ and 5′-CTAGA GCGGC CGCTA CTAGA GTAAA TGTTC-3′. Then the PCR product was digested with *BspH*I and *Not*I, and inserted into the *Nco*I-*Not*I site of the Rh(-1084) plasmid[28]. The *rho:bsx* plasmid was linearized by *Sac*I digestion, and dissolved at a concentration of 25 ng/μl in distilled water containing 0.05% phenol red. The linearized DNA solution was micro-injected into the cell body of one-cell stage embryos.

**Luciferase reporter assay**. Full-length coding sequences of the zebrafish *bsx*, *otx5*, and *crx* were inserted into pcDNA3.1/V5-His-TOPO expression vector (Invitrogen). To generate an *exorh* promoter-reporter construct, the 147-bp fragment upstream from the *exorh* translation initiation site was amplified by PCR with expro147-Fw (5′-TCTCT GGATT ATCCC CCTGT C-3′) and expro147-Rv (5′-GATGG AGAAG TGGAC GATCG-3′) primers from the template plasmid, and inserted into the SmaI site of pGL3-Basic vector (Promega). Human embryonic kidney 293 T cells were plated at $4 \times 10^5$ cells per well in 12-well plates, and transfected with 250 ng of the reporter construct, 250 ng of each expression construct and 0.5 ng of pRL-CMV internal control vector (Promega) using Lipofectamine 2000 reagent (Invitrogen). The total amount of transfected DNA was adjusted to 750 ng/well by mixing with pcDNA3.1/V5-His empty vector. Luciferase reporter activity was measured 24 h after the transfection by using Dual-Luciferase Reporter Assay System (Promega), although the firefly luciferase activity was not normalized by *Renilla* luciferase activity.

**Cell preparation and fluorescence-activated cell sorting**. The pineal gland and the retina were dissected from adult fish of the Tg(*exorh:egfp*) and Tg(*rho:egfp*) transgenic lines, respectively. These tissues were digested in 0.25% trypsin/1 mM EDTA in phosphate-buffered saline (0.8% NaCl, 0.02% KCl, 20 mM phosphate, pH 7.3) at 37 °C for 30 min, and then the reaction was stopped by addition of an equal volume of phosphate-buffered saline containing 2% (w/v) soybean trypsin inhibitor and 20% (v/v) fetal bovine serum. After filtration through a 35-μm nylon mesh, the dissociated cells were sorted by using EPICS ELITE ESP cytometer equipped with Expo32 software (Beckman Coulter). Forward scatter, side scatter and 525-nm green fluorescence intensity were monitored for gating. Sorted cells were directly collected in 1.5-ml microtubes containing TRIzol reagent (Invitrogen) for subsequent extraction of total RNA.

**Ordered differential display**. Ordered differential display was carried out as described[30] with several modifications. For each sample, total RNA (~45 ng) was

used for the synthesis of double-strand cDNA by using Superscript Plasmid System (Invitrogen) with 0.2 μM of T-primer (5′-CGCAG TCGAC CGTTT TTTTT TTTTT-3′). The cDNA samples were digested by *Rsa*I for 2 h at 37 °C and then ligated with adapters (a mixture of two oligonucleotides, 5′-TGTAG CGTGA AGACG ACAGA AAGGG CGTGG TGCGG AGGGG GGT-3′ and 5′-ACCGC CCTCC G-3′) by using Ligation Kit Ver.2 (TaKaRa Shuzo, Kyoto, Japan). Using the cDNA sample as a template, first PCR amplification was performed in PC2-buffer [50 mM Tricine-KOH, 16 mM (NH$_4$)$_2$SO$_4$, 3 mM MgCl$_2$, 150 μg/ml bovine serum albumin (BSA), pH 8.7 at 20 °C] with 250 μM dNTPs, 0.1 μM of each DAd-primer (5′-TGTAG CGTGA AGACG ACAGA A-3′) and T-primer, and 6.25 units KlenTaq1 polymerase (Ab peptides, St. Louis, MO) in a total volume of 25 μl. The amplification reaction was composed of 18 cycles of 94 °C for 40 s, 65 °C for 30 s, and 72 °C for 90 s. The PCR products were then diluted 25-fold in QT buffer (10 mM Tris-HCl, pH 8.0), and the aliquot (1 μl) was used as a template for the second PCR amplification in combination with a pair of two-base-anchored primers, 5′-GCGTG GTGCG GAGGG CGGT(G/T) CNN-3′ and 5′-CGCAG TCGAC CGTTT TTTTT TTTTT (A/C/G)N-3′, in which N represents any base. The combination of the former (16 variants) and the latter (12 variants) primers yields 192 subsets of the amplified products. Amplification reaction was performed in PC2-ODD buffer [20 mM Tricine-KOH, 16 mM (NH$_4$)$_2$SO$_4$, 2.5 mM MgCl$_2$, 150 μg/ml BSA, pH 8.7 at 20 °C] with 250 μM dNTPs, 0.2 μM each of the primers and 2.5 units KlenTaq1 polymerase in a volume of 10 μl for 25 cycles. Then the entire reaction mixture was subjected to 7.5% native PAGE, and the PCR product bands stained with SYBR Green I (Molecular Probes) were quantified by FLA-2000 Bioimage analyzer (Fuji Film, Tokyo, Japan). The selected bands were cut out from the gel and the DNA was eluted from the gel pieces by incubation in 50 μl of QT buffer for 2 h at 55 °C. The eluted DNA was re-amplified by PCR with a pair of Ad2-primer (5′-GCGTG GTGCG GAGGG CGGT-3′) and T-primer, and then the products were separated by agarose gel electrophoresis and purified by phenol extraction and ethanol precipitation. Nucleotide sequences of the purified DNA fragments were determined by direct sequencing with Ad2- and T-primer.

**Selection and amplification binding (SAAB) assay**. A full-length coding sequence of the zebrafish *bsx* was obtained by PCR amplification with a pair of primers (5′-CCCGG AATTC ATGAA TCTGA ACTAC ACG-3′ and 5′-CTGCA GGTCG ACTAG AGTAA ATGTT CAGG-3′) and the *bsx* cDNA plasmid as a template. The amplified fragment was digested with *EcoR*I and *Sal*I, and inserted into *EcoR*I-*Sal*I site of pGEX-5X-1 vector (GE Healthcare). Recombinant GST-Bsx protein was expressed in *Escherichia coli* BL21 strain and purified by affinity chromatography using glutathione-Sepharose 4B resin (GE Healthcare). GST-Bsx protein bound to the resin was washed three times with a Mg$^{2+}$- and ATP-containing buffer (50 mM Tris-HCl, 10 mM ATP, 10 mM MgSO$_4$, pH 7.5) at room temperature to eliminate contaminating proteins such as bacterial chaperon proteins. After overnight elution with 50 mM Tris-HCl (pH 8.0) containing 5 mM glutathione (reduced form) at 4 °C, excess glutathione was removed by gel filtration with a PD-10 column (GE Healthcare). GST protein for control experiments was similarly prepared.

SAAB experiments based on the GST pull-down method were performed as described previously[45] with modifications. For the first round of binding sequence selection, we used double-stranded random oligonucleotides (5′-GAGTC CAGCG AATTC TGTCG-N$_{20}$-GAGTC CTCGA GAGTG TCAAC-3′) having a pair of 20-bp invariant flanking regions[46], which enable PCR amplifications under more stringent conditions and hence prevent concatemerization of the PCR products. The random oligonucleotides (500 ng) were incubated with GST-Bsx (500 ng) or GST (263 ng) protein pre-bound to glutathione-Sepharose 4B beads (5 μl bed volume) in 500 μl of SAAB binding buffer (20 mM Tris-HCl, 50 mM KCl, 0.5 mM EDTA, 1 mM DTT, 50% glycerol, pH 8.0) containing 20 μg/ml bovine serum albumin and 2 μg/ml poly(dI-dC) at 4 °C for 1 h. The pelleted beads were collected by centrifugation at $500 \times g$ for 1 min, washed three times with 1 ml of the SAAB binding buffer, and then resuspended in 30 μl of distilled water. A 10 μl aliquot of the suspension was used as a template for PCR amplification by AmpliTaq Gold DNA polymerase (Applied Biosystems) with a pair of primers, 5′-GTTGA CACTC TCGAG GACTC-3′ and 5′-GAGTC CAGCG AATTC TGTCG-3′, in a 25 μl total reaction volume. The cycling protocol of the PCR reaction was 30 cycles of 94 °C for 30 s and 68 °C for 30 s. A 10-μl aliquot of the PCR product was used for the subsequent round of selection. After five repetitive rounds, the amplified products were cloned into the pCR2.1-TOPO vector and sequenced. Among 89 sequences obtained, eight sequences showing far lower similarities to the others were eliminated from the determination of the consensus binding sequence. Two sequences having multiple potential binding sites were also eliminated.

**Electrophoretic mobility shift assay (EMSA)**. EMSA was performed by using DIG Gel Shift Kit (Roche Diagnostics) according to the manufacturer's instructions. Short double-stranded oligonucleotides for BSXRE probe, BSXRE-mut probe or the competitors were prepared by annealing a pair of synthetic oligonucleotides. Native or mutated DNA probe corresponding to the 147-bp *exorh* promoter was obtained by PCR amplification using a template of the plasmid containing the promoter sequence with or without introduction of mutations. Mutations of P3 and P4 sites (TAATCTG to TACGATG) were introduced into the plasmid having a 1.1-kbp sequence of the *exorh* promoter by using QuikChange Site-Directed

Mutagenesis Kit (Stratagene). Probes without P1 mutation were amplified with expro147-Fw (5′-TCTCT GGATT ATCCC CCTGT C-3′) and expro147-Rv (5′-GATGG AGAAG TGGAC GATCG-3′) primers, while those carrying the P1 mutation were obtained by using expro147-mut-Fw (5′-TCTCT ATCGT ATCCC CCTGT CTG-3′) instead of expro147-Fw. Then the PCR products were purified with QIAquick PCR Purification Kit (QIAGEN) and subjected to digoxigenin-labeling procedures. For binding reactions, 2.0 pmol of GST-Bsx (107 ng) or GST (56 ng) was incubated with 32 fmol of DIG-labeled oligonucleotides or 8 fmol of DIG-labeled DNA probes.

**Statistics and reproducibility**. The statistical data reported include results from at least three biological replicates, whose numbers are stated in each figure. Statistical analyses (Welch's two-sided *t*-test, Games-Howell two-sided test, and Fisher's two-sided exact test) were performed in R (Figs. 2, 4, Supplementary Fig. 3) or Microsoft Excel (Supplementary Figs. 5, 6).

**Reporting summary**. Further information on research design is available in the Nature Research Reporting Summary linked to this article.

## Data availability

All the data are available on request from the authors.

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

## Acknowledgements

We thank Dr. H. Yokoyama (Graduate School of Agricultural and Life Sciences, The University of Tokyo) for fluorescence-activated cell sorting analysis, Drs. M. Matz, S. Lukyanov (Shemyakin and Ovchinnikov Institute of Bioorganic Chemistry), Y. Hirate

and H. Okamoto (Brain Science Institute, RIKEN) for providing the protocols of ordered differential display, Dr. J.O. Liang (University of Minnesota Duluth) for providing otx5 cDNA clone, Dr. M.J. Parsons (Johns Hopkins University) for providing NTR-mCherry plasmid, and Dr. W. Driever (Albert Ludwigs University Freiburg) for providing $bsx^{m1376}$ zebrafish line. We especially thank M. Nagata, W. Qu, C. Tazuke, M. Kamiyama, and S. Matsumoto in our laboratory for technical assistance. This study was supported in part by JSPS KAKENHI Grant Numbers JP19K06758 (to D.K.), JP24227001 (to Y.F.) and JP17H06096 (to Y.F.) and also by research grants from The Naito Foundation (to Y.F.) and from Research Foundation for Opto-Science and Technology (to D.K.). H.M. and Y.A. were supported by Fellowships from the Japan Society for the Promotion of Science for Young Scientists.

## Author contributions

H.M., D.K., and Y.F. designed the research and wrote the paper. H.M., Y.A., and D.K. performed experiments, acquired data, and analyzed the data. All authors have read, commented on, and approved the final paper.

## Competing interests

The authors declare no competing interests.
