## [Peer Review File · Communications Biology]

Reviewers' comments:

Reviewer #1 (Remarks to the Author):

The manuscript "Brain-specific homeobox Bsx specifies identity of pineal gland between serially homologous photoreceptive organs in zebrafish" by Mano et al. reports convincing evidence that the Bsx transcription factor is involved in conferring pineal identity, and inhibiting retina identity, in the pineal complex of zebrafish. This is a very well-constructed study with a set of assays that define the role of Bsx through genetic experiments and promoter analysis. I predict it will be of high interest. However, there is one major concern that needs to be addressed.

This concern is the use of morpholinos for the characterization of the loss-of-function phenotypes of *bsx* (Figure 2) and for the experiments on the relationship of *Ito* other genes involved in pineal complex development (Figure 3). There are now a large group of papers that demonstrate that morpholinos have off target effects (for instance, see Kok et al., 2015, *Developmental Cell* 32: 97-108). Because of this, when using a morpholino in zebrafish, it is necessary to have a loss-of-function mutant in the same gene in order to verify the phenotype.

As the authors themselves explain (Discussion, p. 12) that in all but one case, this is resolved as a recently published paper (Schredelseker and Driever, 2018. *Development* 145: 1-13) used loss-of-function mutant and obtained the same results as in this manuscript. However, there is one main discrepancy that still needs to be resolved. This is also discussed by the authors on p. 12. In this manuscript, loss of *bsx* causes a loss in the expression of the pineal-specific opsin gene *exorh* as well as loss of the *Tg(exorh::egfp)* transgene (Figure 2). In contrast, Schredelseker and Driever found an upregulation of *exorh* mRNA in their *bsx* mutant.

This manuscript provides several other lines of evidence that Bsx is a positive regulator of *exorh*. Most importantly, ectopic expression of Bsx and Otx5 together caused ectopic transcription of the *exorh* in a cell line and in vivo (Figure 4). Thus, in this case, the cumulative evidence suggests that in this case the analysis using the MO, not the mutant, is more likely to be correct. Since the role of *bsx* in positively regulating pineal genes is central to the model of this paper, this needs to be resolved. The most probable explanation is the use of different probes for in situ hybridization (as suggested by the authors on p. 12). I suggest that the authors do a simple experiment comparing the expression pattern of *exorh* mRNA in control and *bsx* MO injected fish using their probe to the 3' UTR of the *exorh* gene and using the probe to the *exorh* coding region used by Schredelseker and Driever. If the issue is just cross reaction of the Schredelseker and Driever probe to other opsin genes, as seems likely, then this issue is resolved.

Reviewer #2 (Remarks to the Author):

In this study, Mano and colleagues address an issue of general interest concerning the molecular mechanisms underlying the unique features that characterize two very homologous biological structures. In particular, focusing on the retina and the pineal organ in zebrafish, they identify *bsx* as a crucial factor able to determine the pineal photoreceptor fate through the genetic interaction with Otx5, but not with Crx.

Although the role of *bsx* in zebrafish pineal organ has been recently analyzed by Schredelseker and Driever (Development 145, dev.163477, 2018), the main findings of this study are novel and, in general, the work is technically sound. However, some issues need to be addressed before the paper can be recommended for publication.

Major points

- The specificity of morpholino-mediated knockdown has been recently debated. In particular, because of some discrepancy observed between the phenotypes of *bsx* morphants and *bsx* mutants, it is necessary to more convincingly show that the effects generated by the *bsx* morpholino oligo used in this study are specific. For instance, the authors could test whether a second *Bsx* morpholino generates the same phenotypes or whether pineal overexpression of *bsx* could rescue the described *bsx* morpholino effects. Furthermore, I wonder whether there is any paralogue of *bsx*, and, if this is the case, whether the *bsx* morpholino used by the authors is specific for just one of the paralogues or is affecting all of them.
- A piece of information is missing to complete the epistatic analysis between *bsx*, *otx5* and *flh* shown in figure 3. The authors show that *flh* is required for *Bsx* expression, suggesting that *flh* acts upstream of *Bsx*, but does *bsx* play any role in the maintenance of *flh* expression? In other words, is *flh* expression affected in *bsx* Mo embryos?

Reviewer #3 (Remarks to the Author):

The manuscript by Mano et al. entitled "Brain-specific homeobox *Bsx* specifies identity of pineal gland between serially homologous photoreceptive organs in zebrafish" show in a very elegant manner a distinct mechanism mediated by *Bsx*. This pathway may represent one of the few examples in the pineal gland and retina fields that distinguishes between these two serially homologous tissues. Thus, it may be very welcome in our community. The manuscript is well organized and written. The authors show the need of *Bsx* for specification of zebrafish pineal identity and therefore, for the identity of both pineal photoreceptor neurons (PRN) and pineal projection neurons. *Bsx* knock-down impaired pineal development and induced retina-specific genes in the pineal gland. In the PRN, *Bsx* regulates the expression of pineal-specific genes such as *exorh* in collaboration with *Otx5*. *Exorh* gene regulation involves the binding of both transcription factors *Bsx* and *Otx5* to PIRE sites and, according to the authors, it might require a still unknown partner over the neighboring PIPE consensus sequence. The experiments were well designed, and the figures tell the story by themselves.

Response to Reviewer#1:

Thank you for reviewing our manuscript. We appreciate your comments. We have taken into consideration all of the reviewers' comments and suggestions into the revision of the manuscript. In order to answer the major criticisms, we performed a series of experiments and obtained data that helped us to strengthen the main conclusion. Our responses are following your comment on the original manuscript as shown below.

Comment 1:

The manuscript “Brain-specific homeobox Bsx specifies identity of pineal gland between serially homologous photoreceptive organs in zebrafish” by Mano et al. reports convincing evidence that the Bsx transcription factor is involved in conferring pineal identity, and inhibiting retina identity, in the pineal complex of zebrafish. This is a very well-constructed study with a set of assays that define the role of Bsx through genetic experiments and promotor analysis. I predict it will be of high interest.

Answer: Thank you for providing a high evaluation on our original manuscript.

Comment 2:

However, there is one major concern that needs to be addressed. This concern is the use of morpholinos for the characterization of the loss-of-function phenotypes of bsx (Figure 2) and for the experiments on the relationship of Ito other genes involved in pineal complex development (Figure 3). There are now a large group of papers that demonstrate that morpholinos have off target effects (for instance, see Kok et al., 2015, Developmental Cell 32: 97-108). Because of this, when using a morpholino in zebrafish, it is necessary to have a loss-of-function mutant in the same gene in order to verify the phenotype.

As the authors themselves explain (Discussion, p. 12) that in all but one case, this is resolved as a recently published paper (Schredelseker and Driever, 2018. Development 145: 1-13) used loss-of-function mutant and obtained the same results as in this manuscript. However, there is one main discrepancy that still needs to be resolved. This is also discussed by the authors on p. 12. In this manuscript, loss of bsx causes a loss in the expression of the pineal-specific opsin gene exorh as well as loss of the Tg(exorh::egfp) transgene (Figure 2). In contrast, Schredelseker and Driever found an upregulation of exorh mRNA in their bsx mutant.

This manuscript provides several other lines of evidence that Bsx is a positive regulator of exorh. Most importantly, ectopic expression of Bsx and Otx5 together caused ectopic transcription of the exorh in a cell line and in vivo (Figure 4). Thus, in this case, the cumulative evidence suggests that in this case the analysis using the MO, not the mutant, is more likely to be correct. Since the role of bsx in positively regulating pineal genes is central to the model of this paper, this needs to be resolved. The most probable explanation is the use of different probes for in situ hybridization (as suggested by the authors on p. 12). I suggest that the authors do a simple experiment comparing the expression pattern of exorh mRNA in control and bsx MO injected fish using their probe to the 3' UTR of the exorh gene and using the probe to the exorh coding region used by Schredelseker and Driever. If the issue is just cross reaction of the Schredelseker and Driever probe to other opsin genes, as seems likely, then this issue is resolved.

Answer: In response to your comment and a similar comment from reviewer #2, we examined pineal *exorh* expression in the *bsx* KO mutant, which Dr. Driever kindly provided us. We thought that the experiment on the KO mutant would provide us a more direct answer to this important question. Then, we found that the *exorh* expression is down-regulated in the *bsx* KO mutant, not only in our *in situ* hybridization experiment (new Fig. S4) but also in the quantitative PCR experiment (new Fig. S5). These results are consistent with our original observation in the *bsx* knock-down experiments. Furthermore, we designed another antisense morpholino, e1i1 MO, by which we verified the *exorh* phenotype of *bsx* morphants in the original manuscript (new Fig. S6). These results together indicate that the effects caused by the first *bsx* MO used in this study are specific. We have included these observations in the revised manuscript (page 7 lines 18-20). We also clarified our statement on the possible reason for the discrepancy in the *exorh* phenotype between Driever's study (ref. 34) and ours as follows:

"Such an apparent difference in exorh expression level might be derived from different designs in the exorh cRNA probes of the in situ hybridization experiments. The previous study³⁴ employed an exorh probe for its protein-coding sequence, whereas we designed it for 3' non-coding sequence unique to exorh mRNA (see Materials and methods for details) to avoid possible cross-reaction to its closest homologue, rho. In fact, the bsx KO mutant showed a reduced level of exorh expression not only in our in situ hybridization experiment (Fig. S4) but also in the quantitative PCR experiment (Fig. S5)." (page 12 line 20 to page 13 line 2).

Response to Reviewer#2:

Thank you for reviewing our manuscript. We appreciate your comments. We have taken into consideration all of the reviewers' comments and suggestions into the revision of the manuscript. In order to answer the major criticisms, we performed a series of experiments and obtained data that helped us to strengthen the main conclusion. Our responses are following your comment on the original manuscript as shown below.

Comment 1:

*The specificity of morpholino-mediated knockdown has been recently debated. In particular, because of some discrepancy observed between the phenotypes of *bsx* morphants and *bsx* mutants, it is necessary to more convincingly show that the effects generated by the *bsx* morpholino oligo used in this study are specific. For instance, the authors could test whether a second *Bsx* morpholino generates the same phenotypes or whether pineal overexpression of *bsx* could rescue the described *bsx* morpholino effects.*

Answer: In response to your comment and a similar comment from reviewer #1, we examined pineal *exorh* expression in the *bsx* KO mutant, which Dr. Driever kindly provided us. We thought that the experiment on the KO mutant would provide us a more direct answer to this important question. Then, we found that the *exorh* expression is down-regulated in the *bsx* KO mutant, not only in our *in situ* hybridization experiment (new Fig. S4) but also in the quantitative PCR experiment (new Fig. S5). These results are consistent with our original observation in the *bsx* knock-down experiments. Furthermore, according to your suggestion, we designed another antisense morpholino, e1i1 MO, by which we verified the *exorh* phenotype of *bsx* morphants in the original manuscript (new Fig. S6). These results together indicate that the effects caused by the first *bsx* MO used in this study are specific. We have included these observations in the revised manuscript (page 7 lines 18-20). We also clarified our statement on the possible reason for the discrepancy in the *exorh* phenotype between Driever's study (ref. 34) and ours as follows:

*"Such an apparent difference in *exorh* expression level might be derived from different designs in the *exorh* cRNA probes of the *in situ* hybridization experiments. The previous study³⁴ employed an *exorh* probe for its protein-coding sequence, whereas we designed it for 3' non-coding sequence unique to *exorh* mRNA (see Materials and methods for details) to avoid possible cross-reaction to its closest homologue, *rho*. In fact, the *bsx* KO mutant showed a reduced level of *exorh* expression not only in our *in situ* hybridization experiment (Fig. S4) but also in the quantitative PCR experiment (Fig. S5)."* (page 12 line 20 to page 13 line 2).

Comment 2:

*Furthermore, I wonder whether there is any paralogue of *bsx*, and, if this is the case, whether the *bsx* morpholino used by the authors is specific for just one of the homologues or is affecting all of them.*

Answer: We sought *bsx* paralogue(s) in the zebrafish DNA/RNA databases by tBLASTn using zebrafish and mouse *Bsx* as queries. We only found distantly related sequences such as *Barx1* and *Barx2*, which do not appear to have function similar to *Bsx*. As we answered to Comment 1, our additional experiments showed that the effects of the *bsx* MO are specific and similar to those of *bsx* KO.

Comment 3:

*A piece of information is missing to complete the epistatic analysis between *bsx*, *otx5* and *flh* shown in figure 3. The authors show that *flh* is required for *Bsx* expression, suggesting that *flh* acts upstream of *Bsx*, but does *bsx* play any role in the maintenance of *flh* expression? In other words, is *flh* expression affected in *bsx* Mo embryos?*

Answer: In response to your comment, we examined *flh* expression in *Bsx*-depleted embryos by quantitative PCR experiments (new Fig. S5, panels C and F). The results showed that *Bsx* depletion had no significant effect (by *bsx* knock-out) or only a marginal effect (by MO-based knock-down) on *flh* expression at 48 hpf. It is thus unlikely that *Bsx* plays a major role in the maintenance of *flh* expression at this stage. We have included these observations in the revised manuscript (page 8 lines 11-12).

Response to Reviewer#3:

Thank you very much for reviewing our manuscript.

Comment:

The manuscript by Mano et al. entitled “Brain-specific homeobox Bsx specifies identity of pineal gland between serially homologous photoreceptive organs in zebrafish” show in a very elegant manner a distinct mechanism mediated by Bsx. This pathway may represent one of the few examples in the pineal gland and retina fields that distinguishes between these two serially homologous tissues. Thus, it may be very welcome in our community. The manuscript is well organized and written. The authors show the need of Bsx for specification of zebrafish pineal identity and therefore, for the identity of both pineal photoreceptor neurons (PRN) and pineal projection neurons. Bsx knock-down impaired pineal development and induced retina-specific genes in the pineal gland. In the PRN, Bsx regulates the expression of pineal-specific genes such as exorh in collaboration with Otx5. Exorh gene regulation involves the binding of both transcription factors Bsx and Otx5 to PIRE sites and, according to the authors, it might require a still unknown partner over the neighboring PIPE consensus sequence. The experiments were well designed, and the figures tell the story by themselves.

Thank you very much for providing a high evaluation on our manuscript.

REVIEWERS' COMMENTS:

Reviewer #1 (Remarks to the Author):

All of my concerns have been addressed in the revised manuscript. It is now acceptable for publication with no additional revisions needed.

Reviewer #2 (Remarks to the Author):

In this new version of the manuscript, all the issues raised in my previous review have been satisfactorily addressed and the paper has significantly improved.